# Characterization of an Extensively Drug-Resistant *Salmonella* Kentucky ST198 Co-Harboring *cfr*, *mcr-1* and *tet*(A) Variant from Retail Chicken Meat in Shanghai, China

**DOI:** 10.3390/foods14173025

**Published:** 2025-08-28

**Authors:** Zeqiang Zhan, Zifeng Mai, Mengjun Hu

**Affiliations:** NMPA Key Laboratorya for Testing Technology of Pharmaceutical Microbiology, MOST-USDA Joint Research Center for Food Safety, Department of Food Science & Technology, School of Agriculture & Biology, Shanghai Jiao Tong University, Shanghai 200240, China

**Keywords:** XDR *Salmonella*, antimicrobial resistance genes, *cfr*, whole genome sequencing, conjugative plasmid, mobile genetic elements, poultry meat

## Abstract

The emergence of extensively drug-resistant (XDR) foodborne pathogens poses grave threats to food safety. This study characterizes the genome of an XDR *Salmonella* Kentucky isolate (Sal23C1) co-harboring *cfr*, *mcr-1* and *tet*(A) from Shanghai chicken meat in 2022, which was the only isolate co-harboring these three key resistance genes among 502 screened *Salmonella* isolates. Genomic analysis revealed that the multidrug resistance gene *cfr*, which confers resistance to phenicols, lincosamides, oxazolidinones, pleuromutilins and streptogramin A, was identified within a Tn*3*-IS*6*-*cfr*-IS*6* structure on the transferable plasmid p3Sal23C1 (32,387 bp), showing high similarity to the *Citrobacter braakii* plasmid pCE32-2 (99% coverage, 99.98% identity). Concurrently, the *mcr-1* gene resided in a *pap2*-*mcr-1* structure on the transferable IncI2 plasmid p2Sal23C1 (63,103 bp). Notably, both genes could be co-transferred to recipient bacteria via conjugative plasmids at frequencies of (1.15 ± 0.98) × 10^−6^. Furthermore, a novel ~79 kb multidrug resistance region (MRR) chromosomally inserted at the *bcfH* locus was identified, carrying *fosA3*, *mph*(A), *rmtB*, *qnrS1* and *bla*_CTX-M-55_. Additionally, a novel *Salmonella* Genomic Island 1 variant (SGI1-KI) harbored *aadA7*, *qacEΔ1*, *sul1* and the *tet*(A) variant. The acquisition of these antibiotic resistance genes in this isolate enhanced bacterial resistance to 21 antimicrobials, including resistance to the critical last-resort antibiotics tigecycline and colistin, which left virtually no treatment options for potential infections. Taken together, this is the first comprehensive genomic report of an XDR poultry-derived *Salmonella* Kentucky isolate co-harboring *cfr*, *mcr-1* and the *tet*(A) variant. The mobility of these resistance genes, facilitated by IS*6* elements and conjugative plasmids, underscores significant public health risks associated with such isolates in the food chain.

## 1. Introduction

*Salmonella* is a primary contributor to foodborne illness globally, affecting both developing and developed countries [1]. Annually, this pathogen is estimated to cause approximately 93.8 million human infections and 155,000 deaths worldwide, underscoring its status as a critical public health challenge [2]. Human salmonellosis typically results from consuming contaminated foods, with poultry, pork and eggs frequently implicated as major sources [2,3,4].

Antimicrobials are widely administered for therapeutic purposes in humans and within food animal production systems [5]. However, evidence indicates that antimicrobial use in food animals promotes the emergence of resistance in *Salmonella* strains transmitted through food [5,6]. Notably, previous research has consistently identified multidrug-resistant (MDR) isolates of *Salmonella* Kentucky (*S.* Kentucky) [7,8]. Recently, the trajectory of MDR development in significant bacterial pathogens, including *Escherichia coli* (*E. coli*), *Klebsiella pneumoniae* (*K. pneumoniae*), *Acinetobacter baumannii*, and *Pseudomonas aeruginosa*, shows a progression towards extensively drug-resistant (XDR) profiles [9,10,11]. According to definitions established by the US Centers for Disease Control and Prevention (US CDC) and the European CDC, XDR refers to isolates resistant to at least one agent in all but two or fewer antimicrobial categories [12]. The presence of XDR bacteria has become an emerging concern, with detections reported in multiple nations such as China, Bangladesh, Egypt, Italy and Pakistan in recent years [8,10,11,13,14,15].

Colistin and tigecycline are critical last-resort antibiotics for treating serious infections caused by XDR Gram-negative pathogens, particularly carbapenemase-producing *Enterobacteriaceae* and *Acinetobacter* [16,17]. However, escalating resistance to both agents in *Enterobacteriaceae* is now evident [16,18], creating a therapeutic crisis for invasive XDR infections. In *Salmonella*, antibiotic resistance typically develops through chromosomal mutations or via lateral genetic exchange mediated by mobile genetic elements (MGEs) like plasmids and transposons (Tns), which can sometimes originate from other bacteria [19]. Among known resistance determinants, the *cfr*, *mcr-1* and *tet*(A) variant genes are particularly significant due to their ability to confer resistance to crucial last-resort agents, including colistin and tigecycline [15,16,20].

Functionally, the *cfr* gene encodes an rRNA methyltransferase conferring cross-resistance to five antimicrobial classes: streptogramin A, pleuromutilins, oxazolidinones, lincosamides and phenicols, while also reducing the efficacy of 16-membered macrolides [20]. Initially identified on the plasmid pSCFS1 in *Staphylococcus sciuri*, *cfr* has since disseminated widely via plasmids/MGEs across diverse Gram-positive and Gram-negative genera [21,22]. In parallel, *mcr-1* encodes a phosphoethanolamine transferase mediating plasmid-borne colistin resistance. Since its 2015 discovery, *mcr-1* has spread globally (>40 countries), with ten variants (*mcr-1* to *mcr-10*) identified in diverse sources [23,24]. Its plasmid association facilitates inter-species spread. Detection is more common in animal-derived isolates than human clinical samples, likely reflecting exposure differences [24,25,26]. As far as we know, only one report describes chromosomal *mcr-1* in *S.* Kentucky from chicken, and plasmid carriage in *Salmonella* remains unreported [27]. Distinctly, the *tet*(A) variant is an intrinsic efflux pump conferring tigecycline resistance in *K. pneumoniae* and *Salmonella*, often synergizing with *ramA* mutations [27,28]. It typically causes low-level resistance and is found on plasmids or chromosomally within SGI1, enabling both horizontal transfer and vertical inheritance.

Previous investigations identified retail food as a major reservoir of *Salmonella* in China, with most food-derived isolates exhibiting high antibiotic resistance [3,4]. To elucidate the mechanisms driving XDR in these isolates, we screened 502 *Salmonella* isolates from Shanghai food samples for key antibiotic resistance genes (ARGs) (*cfr*, *mcr-1* and the *tet*(A) variant). Notably, a food-associated isolate, Sal23C1, was found to exhibit an XDR profile co-harboring all three genes. This study provides an in-depth characterization of Sal23C1, including antimicrobial susceptibility profiling, the evaluation of the *cfr* and *mcr-1* gene transfer potential, and molecular analysis via whole genome sequencing (WGS).

## 2. Materials and Methods

### 2.1. Bacterial Isolates

From January 2020 to January 2023, a total of 502 *Salmonella* isolates were collected from retail food samples from retail markets and supermarkets in Shanghai, China. The food types included pork (*n* = 170), poultry (*n* = 163), beef (*n* = 60), lamb (*n* = 40), shrimp (*n* = 15), roast meat (*n* = 12), freshwater fish (*n* = 9), steamed buns (*n* = 8), frozen dumplings (*n* = 6), sushi (*n* = 5), fried rice (*n* = 2), and cold vegetable dishes (*n* = 2). *Salmonella* isolation was performed using a modified adaptation of the ISO 6579-1:2017 protocol [29]. Briefly, each sample was aseptically collected, placed in a sterile bag, and transported to the laboratory under refrigeration (4–8 °C) within 8 h. For processing, 25 g of each sample was homogenized with 225 mL of buffered peptone water (Huankai, Guangzhou, China) and incubated at 37 ± 1 °C with shaking (160 rpm) for 4 h. Then, 1 mL of the pre-enrichment culture was transferred to 9 mL of tetrathionate broth (Huankai, China) and incubated at 42 °C. After enrichment, a loopful of culture was streaked onto xylose lysine tergitol 4 agar (XLT-4; Huankai, China). Presumptive *Salmonella* colonies were subcultured on CHROMagar™ *Salmonella* plates (CHROMagar, La Plaine St. Denis, France) and incubated at 37 °C for 24 h. *Salmonella* identification was initially based on key biochemical characteristics, including H_2_S production, anaerobic glucose fermentation without gas production, and urease negativity. Further confirmation was conducted using API 20E test strips (bioMérieux, Marcy-l’Étoile, France), a commercial identification system for Enterobacteriaceae. The verified isolates were subsequently subcultured on Luria–Bertani agar and genetically confirmed through PCR targeting the *invA* gene with specific primers (invA-F: CTTGATTGAAGCCGATGCCG; invA-R: TCATCGCACCGTCAAAGGAA). The identification of the serotype was performed through standardized biochemical assays and commercial antisera (Statens Serum Institute, København, Denmark) following manufacturer protocols [30].

### 2.2. Polymerase Chain Reaction and DNA Sequencing of ARGs

Total genomic DNA from overnight cultures of *Salmonella* was isolated with the TIANamp Bacteria DNA Kit (Tiangen Biotech, Beijing, China). In brief, bacterial cells from 1–2 mL of culture were pelleted by centrifugation (10,000 rpm, 2 min). The pellet was resuspended in 200 μL of Buffer GA, treated with 20 μL of Proteinase K and 220 μL of Buffer GB, mixed, and incubated at 70 °C for 10 min until lysate clarification. After adding 220 μL of ethanol, the mixture was transferred to a CB3 spin column and centrifuged (10,000 rpm, 30 s). The column was washed once with 500 μL of Buffer GD and twice with 600 μL of Buffer PW, with each wash step followed by a 30 s spin. After a final 2 min centrifugation and air-drying for 2–5 min, the DNA was eluted using 50–100 μL of Buffer TE. The DNA quality and concentration were determined via NanoDrop (Thermo Fisher Scientific, Waltham, MA, USA). The detection of the ARGs *cfr*, *mcr-1* and the *tet*(A) variant in the collected isolates was performed using polymerase chain reaction (PCR). The primer sequences utilized were as follows: for the *cfr* gene, forward primer cfr-F: GTGAAGCTCTAGCCAACCGTC and reverse primer cfr-R: GCAGCGTCAATATCAATCCC; for the *mcr-1* gene, forward primer mcr-1-F: GCAACCAAGCCTGATATGCG and reverse primer mcr-1-R: CGCTTAAAATACGCAGGCCC; and for the *tet*(A) variant gene, forward primer tet(A)-F: TCTGGTTCACTCGAACGACG and reverse primer tet(A)-R: AGCCCGTCAGGAAATTGAGG. Positive PCR amplicons were sent to Sangon Biotech (Shanghai) for sequencing confirmation. Isolates found to co-harbor all three ARGs (*cfr*, *mcr-1* and the *tet*(A) variant) are referred to as CMT isolates.

### 2.3. Antimicrobial Susceptibility Testing of CMT Isolates

The *Salmonella* isolate Sal23C1 was identified as a CMT isolate, co-harboring the ARGs *cfr*, *mcr-1* and the *tet*(A) variant. The antimicrobial susceptibility testing of Sal23C1 was performed using agar dilution and broth microdilution methods according to Clinical and Laboratory Standards Institute (CLSI; 2023) guidelines [31]. The agar dilution method was applied to the following antibiotics at the indicated concentration ranges: sulfisoxazole (16–2048 mg/L), trimethoprim–sulfamethoxazole (0.5/9.5–16/304 mg/L), nalidixic acid (8–128 mg/L), ofloxacin (0.5–32 mg/L), ciprofloxacin (0.06–8 mg/L), ampicillin (4–128 mg/L), amoxicillin–clavulanic acid (4/2–128/64 mg/L), cefotaxime (1–64 mg/L), cefepime (1–64 mg/L), tetracycline (4–128 mg/L), chloramphenicol (8–128 mg/L), florfenicol (4–128 mg/L), streptomycin (8–512 mg/L), gentamicin (2–64 mg/L), amikacin (4–256 mg/L), kanamycin (8–256 mg/L), rifampin (4–128 mg/L), azithromycin (4–128 mg/L), fosfomycin (16–512 mg/L) and meropenem (0.06–16 mg/L). For tigecycline and colistin, the MICs were determined using the broth microdilution method. All the antibiotics used in this study were purchased from Sigma-Aldrich (Saint Louis, MO, USA). The minimum inhibitory concentrations (MICs) were determined following CLSI (2023) standards [31], with *Enterococcus faecalis* 29,212 and *E. coli* 25,922 serving as quality control strains.

### 2.4. Whole Genome Sequencing and Analysis of the CMT Isolate

The raw sequence reads from the *Salmonella* isolate Sal23C1 were assembled de novo using the PacBio RS II Analysis system via the HGAP assembler. Leveraging the hierarchical strategy of HGAP along with PacBio long-read sequencing, this approach enabled the production of highly accurate and contiguous genome assemblies—essential for achieving complete genomic sequences. Genomic characterization was subsequently conducted using bioinformatic tools hosted by the Center for Genomic Epidemiology (https://www.genomicepidemiology.org/, accessed on 7 August 2025). Specifically, PlasmidFinder 2.1 (https://cge.food.dtu.dk/services/PlasmidFinder/, accessed on 7 August 2025) was used to identify plasmid replicons, ResFinder 4.1 (https://cge.food.dtu.dk/services/ResFinder/, accessed on 7 August 2025) was used to detect antimicrobial resistance genes, and MLST 2.0 (https://cge.food.dtu.dk/services/MLST/, accessed on 7 August 2025) was used to determine the multi-locus sequence type (MLST). The serotype was predicted using SeqSero2 v1.2.1 (http://www.denglab.info/SeqSero2) [32]. Insertion sequence (IS) elements were identified by querying the ISFinder database (https://isfinder.biotoul.fr/about.php, accessed on 7 August 2025).

### 2.5. Conjugation Transfer Assay

To assess the transferability of the *cfr* and *mcr-1* genes from the *Salmonella* isolate Sal23C1, conjugation experiments were performed using rifampin-resistant *E. coli* J53 as the recipient strain and Sal23C1 as the donor. Selective conditions were established based on differential antibiotic susceptibility: Sal23C1 exhibited no growth on LB agar supplemented with 200 mg/L rifampin, while *E. coli* J53 failed to proliferate on plates containing 2 mg/L colistin plus 16 mg/L florfenicol. The donor and recipient cultures were separately incubated in LB broth with shaking (37 °C, 4–6 h). Subsequently, 1 mL aliquots were centrifuged, and the cell pellets were resuspended, mixed at a 1:1 ratio, and then spot-plated onto LB agar for overnight mating at 37 °C. Following overnight mating at 37 °C, the conjugation mixture underwent serial dilution in phosphate-buffered saline before plating onto triple-antibiotic selection plates (200 mg/L rifampin, 16 mg/L florfenicol, 2 mg/L colistin). The plates were incubated at 37 °C for 24–48 h. Putative transconjugants appearing as single colonies were PCR-verified to exclude false-positive candidates. We calculated the transfer frequency as the number of transconjugants per donor and express the conjugation efficiency as the mean ± standard deviation from three independent experiments.

### 2.6. Phylogenetic Tree Analysis of the S. Kentucky Isolates and IncI2 Plasmids Harboring the mcr-1 Gene

To investigate the phylogenetic relationships among *mcr-1*-positive *S.* Kentucky isolates, we performed core genome phylogeny using the kSNP4.1 pipeline [33]. This analysis incorporated the study isolate Sal23C1 alongside fourteen *S.* Kentucky genomes acquired from public repositories (NCBI and Enterobase). In parallel, we constructed a comparative phylogeny for *mcr-1*-positive IncI2 plasmids, integrating one plasmid identified herein with fifteen publicly available NCBI plasmid sequences. All the resulting phylogenetic trees were visualized using the iTOL v6 platform (https://itol.embl.de, accessed on 7 August 2025). Detailed in Appendix A (‘Isolates and plasmids for phylogenetic analysis’) is a complete inventory of the bacterial isolates and plasmids employed for this study’s phylogenetic examination.

## 3. Results and Discussion

### 3.1. Phenotypic Characteristics and Antibiotic Resistance Determinants of Isolate Sal23C1

The PCR screening of 502 *Salmonella* isolates from retail foods revealed varied prevalence of key resistance genes. The *tet*(A) variant was the most prevalent, detected in 124 isolates (24.7%), followed by *mcr-1*, which was present in 7 isolates (1.4%). The *cfr* gene was the rarest, identified in only one isolate (0.2%). Notably, a single isolate (0.2%), designated Sal23C1 and collected from retail chicken meat in Shanghai in 2022, was found to co-harbor all three genes—*cfr*, *mcr-1* and the *tet*(A) variant. Serotyping and MLST analysis confirmed this isolate Sal23C1 as *S.* Kentucky ST198. Antimicrobial susceptibility testing revealed that Sal23C1 exhibited simultaneous resistance to a broad spectrum of agents, including tetracycline, ciprofloxacin, ofloxacin, nalidixic acid, cefepime, ampicillin, cefotaxime, amoxicillin–clavulanic acid, streptomycin, gentamicin, amikacin, kanamycin, chloramphenicol, florfenicol, azithromycin, fosfomycin, sulfisoxazole, rifampicin and trimethoprim–sulfamethoxazole. This profile meets XDR criteria according to contemporary ECDC/US CDC definitions [12]. Critically, this isolate Sal23C1 also exhibited resistance to the last-line agents colistin and tigecycline, while demonstrating susceptibility only to meropenem among all the tested antibiotics (Table 1).

The assembly of the total reads resulted in four contigs with a total length of 5,079,104 bp and an N50 of 4,889,727, achieving a genome coverage of 250×. The WGS showed that the complete genome of Sal23C1 (GenBank PRJNA1303235) comprises one chromosome and three plasmid replicons, which include the chromosome (Sal23C1, 4,889,727 bp, GC 52.2%), IncI (p1Sal23C1, 93,887 bp), IncI2 (p2Sal23C1mcr, 63,103 bp) and one novel type (p3Sal23C1cfr, 32,387 bp). Genomic analysis identified 20 known ARGs (Table 1), consistent with the phenotypic resistance profile. These included genes conferring resistance to chloramphenicols (*cfr*, *floR*); colistin (*mcr-1*); beta-lactams (*bla*_CTX-M-55_, *bla*_TEM-1B_); rifamycin (*arr-2*); fluoroquinolones (*qnrS1*); aminoglycosides (*rmtB*, *aac*(3)*-IId*, *aac*(6′)*-Iaa*, *aadA7*, *aadA17*, *aph*(3′)*-Ia*); tetracyclines (*tet*(A) variant); folate pathway antagonists (*sul1*, *dfrA14*); lincosamides (*lnu*(F)); fosfomycins (*fosA3*); macrolides (*mph*(A)); and quaternary ammonium compounds (*qacEdelta1*). Additionally, three mutations in the Quinolone Resistance-Determining Region (QRDR) were identified in the chromosome: *parC* (S80I) and *gyrA* (D80N and S83F), known to confer quinolone resistance. This genetic basis fully explains the observed XDR phenotype.

Our findings demonstrated the presence of XDR profiles in the *Salmonella* isolate Sal23C1 collected from retail chicken meat, suggesting a potential animal origin transmitted via the food supply chain, which was aligned with reports of XDR *Salmonella* in animals and patients [34]. More importantly, colistin and tigecycline are regarded as antibiotics of last resort for treating serious clinical infections caused by XDR Gram-negative organisms, particularly carbapenemase-producing *Enterobacteriaceae* and *Acinetobacter* isolates [16,17]. However, the emergence and spread of plasmid-mediated resistance genes like *mcr* (mediated resistance to colistin) or *tet*(A) variants (mediated resistance to tigecycline) significantly impair the therapeutic efficacy of these agents [15,16]. Critically, the co-occurrence of *cfr*, *mcr-1* and the *tet*(A) variant in Sal23C1, marking the first report of these specific genes together in *Salmonella*, directly compromises both colistin and tigecycline. This leaves tigecycline and colistin ineffective, leaving meropenem as the sole tested effective agent for treating potential invasive infections caused by such isolates, creating a critical therapeutic crisis. In addition, the *cfr* gene provides cross-resistance to multiple classes (streptogramin A, pleuromutilins, oxazolidinones, lincosamides, phenicols) [20]. Extended-spectrum β-lactamases (ESBLs) are key β-lactam resistance mechanisms; the co-existence of *bla*_CTX-M-55_ (a CTX-M-15 variant with enhanced cephalosporin hydrolysis due to A80V) and *bla*_TEM-1B_ in Sal23C1 significantly contributes to its β-lactam resistance [35]. The acquired *fosA3* confers high-level fosfomycin resistance, previously seen in animal-source *Salmonella* and human gastroenteritis cases [36]. Furthermore, it was found that the *Salmonella* isolates exhibited resistance to nalidixic acid when they carried only the gyrA/parC mutations. Our research indicates that the presence of *qnrS1* with gyrA/parC mutations enhances resistance to ciprofloxacin, which is observed in *Salmonella* (≥8 μg/mL) [35].

The emergence of the colistin–tigecycline-XDR *S.* Kentucky ST198 in retail chicken meat highlights the critical need for comprehensive intervention strategies within poultry production systems to curb the dissemination of antimicrobial resistance. The extensive arsenal of ARGs, such as *mcr-1* and the *tet*(A) variant, grants strains like Sal23C1 their challenging resistance profile. Alarmingly, antibiotic candidates under development for WHO critical-priority pathogens show concerning cross-resistance rates with existing agents, particularly against such isolates [37]. This underscores the urgent need to elucidate the genomic characteristics of these strains and understand the mechanisms enabling such devastating resistance combinations. To address this threat, antibiotic stewardship programs must be rigorously implemented to reduce the non-therapeutic use of critically important antibiotics, including colistin and tigecycline, in animal husbandry. Restrictions on these agents could mitigate the selective pressure driving the acquisition and spread of resistance genes. Enhanced biosecurity measures are also essential to prevent the introduction and transmission of resistant pathogens within farms. These include strict sanitation protocols, controlled access to poultry houses, and the use of protective clothing and equipment to minimize cross-contamination. Vaccination programs against *Salmonella* and the use of probiotics or prebiotics to promote gut health may further reduce reliance on antibiotics. Furthermore, continuous surveillance of antimicrobial resistance in both animals and food products, coupled with whole genome sequencing, is vital to track resistance trends and identify emerging threats early. The integration of these strategies within a One Health framework is crucial to effectively combat the spread of XDR pathogens from poultry to humans through the food chain.

### 3.2. Phylogenetic Relationship of mcr-1-Positive S. Kentucky Isolates and IncI2 Plasmids

To investigate the global relatedness of the *mcr-1*-positive isolate identified in this study among other *mcr-1*-positive *S*. Kentucky strains, we performed phylogenetic analysis on 15 isolates (our isolate plus 14 *mcr-1*-positive *S*. Kentucky isolates sourced from NCBI and Enterobase). This analysis assessed clonal relationships (Figure 1A). Our findings indicate that *mcr-1*-positive *S*. Kentucky ST198 has been collected from various reservoirs, including humans, animals and food. These isolates grouped into distinct subclades I and II. Notably, within subclade II, isolates derived from humans, pork and poultry in countries such as Morocco, the UK, China and France clustered together, including the isolate from this study. The short branch lengths (scale bar < 0.1) observed among subclade II isolates suggest a close genetic link, despite their diverse geographical and source origins. Specifically, our isolate (Sal23C1) showed the closest phylogenetic affinity to two Chinese pork-derived isolates (Sal_P23040 and Sal_P23041) and a French clinical human isolate (traces-0lhyZjN). Consistent with their phylogenetic relatedness, all 15 *mcr-1*-positive isolates shared resistance determinants for sulfonamides (*sul1*) and quinolones (*gyrA*(S83F) and *parC*(S80I)), which were universally present (100%). Additionally, the tetracycline resistance gene *tet*(A) was detected in 14 isolates (93.3%), and *bla*_TEM-1B_ in 12 isolates (80.0%). Importantly, *bla*_CTX-M-55_ and *qnrS1* were uniquely identified in isolates originating from China. These results suggest the food chain as a potential transmission route for *mcr*-*1*-positive *S*. Kentucky within China.

Concurrently, phylogenetic analysis of 16 *mcr-1*-harboring IncI2 plasmids (Figure 1B) revealed considerable diversity and multiple branches among plasmids obtained from diverse sources (e.g., humans, food) in different countries (e.g., China, Thailand, Oman, Argentina). A significant observation was the close clustering of the plasmid p2492-4-MCR-1 (CP044025.1, 60,688 bp), originating from US pork, with the plasmid p2SalC1mcr identified in the current study.

*Salmonella* demonstrates broad host adaptability, capable of infecting a wide range of animal species, including mammals, birds and insects [38]. Consequently, these broad-host-range isolates can be disseminated via animal feces or through the food chain [38]. A phylogenomic comparison of the isolates characterized here with all the available *mcr-1*-positive *S*. Kentucky sequences from food, human and environmental sources confirmed their close relatedness, forming a distinct cluster alongside a human isolate from France. Furthermore, the minimal genetic divergence observed in the phylogenetic branches aligns with the consistent ST198 assignment. Although in vivo transmission experiments were not performed to definitively trace pathways, these results indicate phylogenetic proximity between chicken-derived and human *S*. Kentucky ST198 isolates. This supports the potential for chicken to act as a significant vector for *mcr-1*-positive *S*. Kentucky ST198 transmission. Enhanced surveillance efforts should therefore prioritize monitoring the IncI2 plasmid-mediated dissemination of *mcr-1* within this lineage.

### 3.3. Molecular Characteristics of the Multidrug Resistance Region and Salmonella Genomic Island 1

Genomic analysis revealed a chromosome length of 4,889,727 bp and a GC content of 52.2% in the studied isolate. Of the 20 identified ARGs, eighteen resided within the multidrug resistance region (MRR) and the novel *Salmonella* Genomic Island 1 variant (designated SGI1-KI) on the chromosome, while only *mcr-1* and *cfr* were located in plasmids. Notably, the insertion of the MRR disrupted the open reading frame (ORF) of the *bcfH* gene in this ST198 isolate Sal23C1, contrasting with the intact *bcfH* observed in ST314 clade reference genomes (e.g., GCF_002952975.1) (Figure 2A). Given that Sal23C1 was collected from chicken meat and that the role of the bovine colonization factor (*bcf*) fimbriae in avian intestinal colonization is minimal, the disruption of *bcfH* by the MRR insertion may confer a significant survival advantage through enhanced antibiotic resistance without substantially compromising fitness in its primary reservoir host [39,40].

This study identified a chromosome-localized *qnrS1* gene, differing from its typical location on IncHI2 plasmids [41]. This chromosomal integration likely promotes *qnrS1′*s stable persistence within the *S.* Kentucky isolate Sal23C1. Additionally, mutations in *parC* (S80I) and *gyrA* (S83F, D87N) were detected. As depicted in Figure 2A, *bla*_CTX-M-55_ was situated within an ~79 kb MRR positioned downstream of the *bcfBCDEFG* gene cluster. The MRR, flanked by IS26 elements integrated into *bcfH*, contained multiple ARGs with the following structure—IS*26-*IS*26-floR-*IS*26-qnrS1-*IS*26-bla*_CTX-M-55_*-bla*_TEM-1_*-*IS*26-fosA3-*IS*26-rmtB-bla*_TEM-1_*-*IS*26-aac*(3)*-IId-*IS*26-lnu*(F)*-aadA17-*IS*26-mph*(A)-IS*26-cmlA-arrr-2-dfrA14-*IS*26-*IS*26-aph*(3′)*-Ia-*IS*26*)—conferring resistance to multiple antimicrobials, including ciprofloxacin, ofloxacin, cefepime, cefotaxime, amoxicillin–clavulanic acid, amikacin, azithromycin and fosfomycin.

The integration of the MRR into the *bcfH* locus, resulting in its disruption, may enhance *S.* Kentucky ST198 survival. Given that ARGs in *Salmonella* often emerge from selective pressures associated with antibiotic use in animal agriculture and veterinary medicine [42], the MRR-mediated antibiotic resistance likely provides a significant survival advantage in intensive antibiotic environments. Although the *fim* operon is the most highly expressed *Salmonella* fimbrial operon, *bcf* expression is specifically induced within the bovine ileum [43] and its role in avian intestinal colonization is minimal [44].

Analysis confirmed the presence of the SGI1-KI in this isolate and, upon comparison with the traditional SGI1-K structure (reference AY463797), identified it as a distinct variant (Figure 2B). SGI1-KI retained the genes *accCA5, aadA7*, *qacEΔ1* and *sul1* and the *tet*(A) variant, but lacked *tnpR*, the segment between S023 and *resG,* IS*1133*, *strAB*, and the *bla*_TEM-1b_ genes [45]. The ST198 isolate Sal23C1 harbored a novel SGI1-K variant incorporating a mercury resistance module. Structural comparisons with the SGI1-K prototype confirmed specific deletions (*tnpR,* S023-*resG* segment, IS*1133*, *strAB*, *bla*_TEM-1b_) [45]. These alterations indicate rapid intraclonal evolution, differing from patterns in other isolates and potentially reflecting localized selection pressures. Previous reports describe SGI1-K in *S.* Kentucky ST198 as highly mosaic, with IS*26*-mediated gene acquisitions or losses generating extensive structural diversity [46]. This inherent genomic flexibility within SGI1-K likely confers adaptive advantages to this high-risk clone. The acquisition of extensive resistance determinants, particularly through large genomic insertions like the MRR that disrupt native genes (e.g., *bcfH*), can potentially impose a fitness cost on the bacterial host in the absence of antibiotic selection. However, the global dissemination and persistence of XDR *S.* Kentucky ST198 clones suggest that any such costs are effectively mitigated. The chromosomal integration of key resistance genes within stable genetic platforms (MRR, SGI1-KI) enhances their persistence compared to plasmid-borne genes, which may be lost more readily. Furthermore, the observed genomic rearrangements and deletions may themselves represent compensatory evolution that reduces the metabolic burden. The formidable survival advantage provided by multidrug resistance in environments with high antibiotic selective pressure appears to far outweigh any potential fitness defects, enabling the successful clonal expansion and spread of this high-risk lineage with minimal trade-offs.

### 3.4. Genomic Characteristics of cfr-Positive Plasmid p3Sal23C1cfr

The isolate Sal23C1 harbored three distinct replicons: an IncI plasmid (p1Sal23C1, 93,887 bp), an IncI2 plasmid (p2Sal23C1mcr, 63,103 bp) and a novel plasmid type (p3Sal23C1cfr, 32,387 bp) carrying the *cfr* gene. The *cfr*-positive plasmid, designated p3Sal23C1, had a total length of 32,387 bp and a GC content of 45.7% (Figure 3A). Comparative analysis revealed that p3Sal23C1 exhibits high similarity (96–100% query coverage; 99.9–100.0% nucleotide identity) to known *cfr*-carrying plasmids from *Citrobacter braakii* (CP138572.1), *Salmonella* (CP116020.1) and *E. coli* (KR779901.1, KP324830.1). The closest match was plasmid pCE32-2 (CP138572.1), with 99% coverage and 99.98% identity, although a 344 bp size difference was noted, and the orientation of the *cfr* gene was reversed in p3Sal23C1.

Collinearity assessment indicated that p3Sal23C1 shares an identical plasmid backbone (95% coverage, 100.0% identity) with plasmid pYUXYEH3314-3 (CP110990.1) from *E. coli*. However, p3Sal23C1 contained an additional 1406 bp segment (Figure 3B). This region comprises the *cfr*-associated module. Blastn analysis demonstrated that the structure IS*6*-*cfr*-IS*6*-Tn*3* within this module is consistent with that found in the plasmid pYUSHP29-3 (CP087283.1) from *Leclercia adecarboxylata*. Notably, IS*6*-like elements (IS*15* family transposases) flank the *cfr* gene in the same orientation, suggesting the potential for mobilization via IS elements to form translocatable *cfr*-carrying units, thereby enabling the spread of linezolid resistance. The plasmid p3Sal23C1cfr encodes proteins associated with conjugative transfer functions (including *traC*, *traD*, *traR*, origin of transfer (*oriT*) *region*, a type IV secretion system (*T4SS*) cluster, and type IV coupling protein (*T4CP*) genes) alongside plasmid partitioning genes (*parA* and *parB*).

The multiple resistance gene *cfr* was first identified in a bovine *Staphylococcus sciuri* isolate from Germany in 2000, with the first human clinical case reported in an MRSA isolate from Colombia in 2005 [21,47]. Our study identifies a *cfr*-carrying *Salmonella* isolate originating from chicken meat in 2022. This finding also highlights foodborne *Salmonella* as a significant reservoir for the *cfr* gene. To our knowledge, this represents the first documented instance of a *cfr*-harboring plasmid within *Salmonella* derived from chicken meat. Although *cfr* was initially and remains predominantly detected in Gram-positive bacteria like *Staphylococcus aureus* and *Staphylococcus capitis*, its increasing identification in Gram-negative species is an alarming trend demanding vigilance [22,48,49].

### 3.5. Genomic Characteristics of mcr-1-Positive Plasmid p2Sal23C1mcr

The colistin resistance gene *mcr-1* was identified on the IncI2 plasmid p2Sal23C1mcr. This plasmid measures 63,103 bp in length and has a GC content of 42.7%. The IncI2 plasmids from the *S*. Kentucky isolate p2Sal23C1mcr encode key components essential for conjugative transfer, including an *oriT* region, a *T4SS* cluster and *T4CP* genes. Comparative genomic analysis (Figure 4A) revealed significant similarity between p2Sal23C1mcr and known *mcr-1*-harboring IncI2 plasmids from *E. coli* (accessions: KX034083.1, MF693349.1, CP044025.1, KY802014.1, APO17614.1, CP075719.1, APO17622.1, APO17619.1, KU761326.1, KX254342.1, KY693674.1, MG825374.1, MW373507.1 and MN689940.1) and *Salmonella* (plasmid p2Sal_P23040), exhibiting 87–95% query coverage and 98.8–99.9% nucleotide identity. Among these, the plasmid pECJS-61-63 (KX254342.1) from *E. coli* showed the closest resemblance (95% coverage, 99.8% identity). Notably, the majority of these IncI2 plasmids solely carry the *mcr-1* resistance gene. Consistent with typical *mcr-1* organization, the gene’s coding sequence is invariably positioned immediately upstream of an open reading frame encoding a PAP2 family protein (Figure 4B). The analysis of specific plasmids (AP017614.1, AP017619.1, AP017622.1, CP075719.1, KX034083.1) detected the complete IS*Apl1* element downstream of the *mcr-1* cassette.

Research classifies the genetic environment of *mcr-1* into four primary structural types [50]. The most prevalent configuration is Tn*6330*, a 2609 bp composite transposon containing the *mcr-1* gene adjacent to a 765 bp ORF predicted to encode a PAP2 superfamily protein. Tn*6330* is typically bounded by two copies of IS*Apl1*, an IS*30* family insertion sequence [51]. In contrast, the *mcr-1*-positive isolate examined here exhibited an *mcr-1-pap2* structure lacking flanking IS*Apl1* elements. This absence points to potential recombination-mediated excision of IS*Apl1* from the *mcr-1* locus during the isolate’s evolution. Such structural simplification aligns with prior findings suggesting that IS*Apl1* loss may enhance the stability and persistence of *mcr-1* within plasmid vectors, thereby facilitating wider dissemination of this critical resistance determinant [52]. Consequently, a deeper understanding of *mcr-1* transmission dynamics necessitates further investigation into its precise transposition mechanisms and evolutionary history.

### 3.6. Conjugation Transfer of Plasmids Carrying mcr-1 and cfr

Conjugation experiments were performed to evaluate the transferability of the plasmids p2Sal23C1mcr (carrying *mcr-1*) and p3Sal23C1cfr (carrying *cfr)* from the donor isolate. The resulting transconjugants grew normally on these selective plates. PCR analysis confirmed that these transconjugants harbored both the *cfr* and *mcr-1* genes, indicating successful co-transfer of the respective plasmids from the donor to the *E. coli* J53 recipient. The conjugation frequency was (1.15 ± 0.98) × 10^−6^. This frequency range indicates that the plasmids are transferable at detectable and biologically relevant rates.

Genetic characterization revealed that both of the plasmids p2Sal23C1mcr and p3Sal23C1cfr harbored intact conjugative modules, including functional *oriT*, *T4SS* clusters, and *T4CP* genes. This finding is consistent with previous research [53], which demonstrated that such modules enable efficient horizontal plasmid transfer between bacterial strains. These conserved genetic elements provide a mechanistic basis for the observed experimental plasmid transfer.

Plasmids are primary vectors for the horizontal dissemination of resistance genes like *cfr* and *mcr-1*. The *cfr* gene has been identified on diverse plasmid types (e.g., pSCFS-like, pBS-like, pSS-like, p004-737×, pERGB, pMSA16), frequently co-harboring additional resistance determinants [22,48]. Similarly, *mcr-1* is commonly associated with plasmids (IncX4, IncI2, IncHI2, IncA, IncI1, IncX1), which also often carry multiple resistance genes [3,16,24,25]. The presence of transferable multi-resistance plasmids harboring *cfr* and *mcr-1* significantly expands the antibiotic resistance spectrum, intensifies co-selective pressures, and promotes the widespread dissemination of these critical resistance genes within foodborne *Salmonella* [3,54]. This poses a substantial threat to food safety and public health. Consistent with this pattern, our study confirmed the plasmid localization of *cfr* and *mcr-1*, underscoring the pivotal role of plasmids as vectors of resistance dissemination in *Salmonella*.

## 4. Conclusions

This study reports the first detection of a *cfr*-producing, colistin- and tigecycline-resistant and XDR *S.* Kentucky ST198 isolate from retail chicken meat, which is a highly concerning finding. Multiple acquired ARGs were identified in the MRR, SGI1-KI and plasmids within this isolate. Critically, the *mcr-1* and *cfr* genes reside on transferable plasmids, facilitating their dissemination across bacterial species. The acquisition of these ARGs confers an XDR phenotype, enabling the pathogen to persist and spread under diverse antibiotic selection pressures, ultimately accelerating the selection of resistant isolates. The convergence of resistance to last-resort antibiotics, mediated by *cfr*, *mcr-1* and the *tet*(A) variant, within this single, globally disseminated, high-risk clone (*S.* Kentucky ST198), facilitated by mobile genetic elements, represents an extreme public health threat. In light of these findings, it is imperative to implement comprehensive intervention strategies within poultry production systems to curb the dissemination of such high-risk resistant bacteria. Antibiotic stewardship programs that enforce the responsible use of antimicrobials—including restrictions on non-therapeutic applications and the prohibition of using critically important antibiotics for growth promotion—are essential to reduce the selective pressure for resistance development. Additionally, enhanced biosecurity measures should be adopted to prevent the introduction and spread of pathogens within farms. These include strict access control, sanitation protocols, effective waste management, and rodent and insect control. Integrated surveillance of antimicrobial resistance across the farm-to-fork continuum should be strengthened, coupled with periodic training of farmers and veterinarians on appropriate antibiotic use and infection prevention practices. Without such multidimensional interventions, the continued emergence and spread of pan-resistant pathogens in the food chain may become increasingly common, posing severe risks to public health.

## Figures and Tables

**Figure 1 foods-14-03025-f001:**
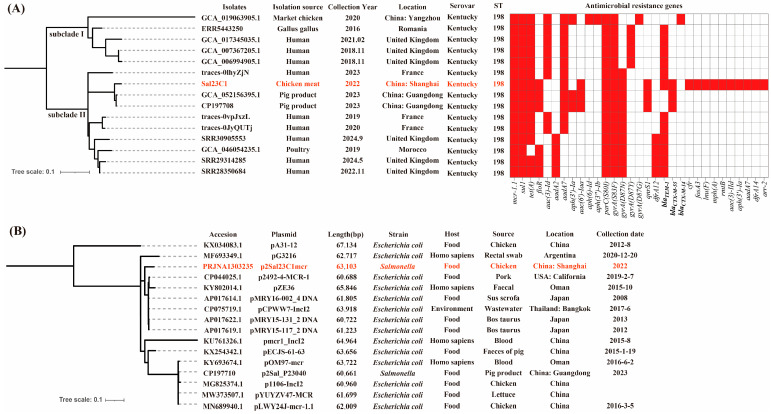
Phylogenetic relationship of *mcr-1*-positive *Salmonella* Kentucky isolates and IncI2 plasmids. (**A**). Phylogenetic analysis of 15 *mcr-1*-positive *S.* Kentucky isolates. This analysis included 14 isolates sourced from NCBI and EnteroBase, plus the isolate sequenced in this study (Sal23C1, highlighted in red). The figure presents the basic isolate information and the presence of antimicrobial resistance genes (indicated in red). (**B**). Phylogenetic analysis of 16 IncI2 plasmids harboring the *mcr-1* gene. This analysis included 15 plasmids sourced from NCBI, plus the plasmid sequenced in this study (p2Sal23C1mcr, highlighted in red).

**Figure 2 foods-14-03025-f002:**
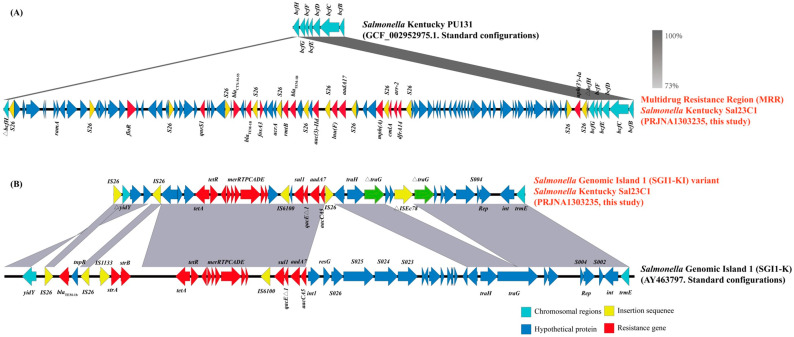
Molecular characteristics of the multidrug resistance region and *Salmonella* Genomic Island 1. (**A**). Comparative analysis of the genetic structure of the multidrug resistance region (MRR) inserted within the *bcfH* gene in *S.* Kentucky ST198 isolate. Chromosomal genes are indicated in light blue, insertion sequences (IS) in yellow, and resistance genes in red. (**B**). Comparative analysis of the genetic structure of the *Salmonella* Genomic Island 1 (SGI1) variant inserted downstream of the *trmE* gene in *S.* Kentucky ST198 isolates, aligned against the reference SGI1 sequence (AY463797). Resistance genes are shown in red, and insertion sequences (IS) in yellow. Regions of sequence similarity identified by BLASTn are indicated by gray shading.

**Figure 3 foods-14-03025-f003:**
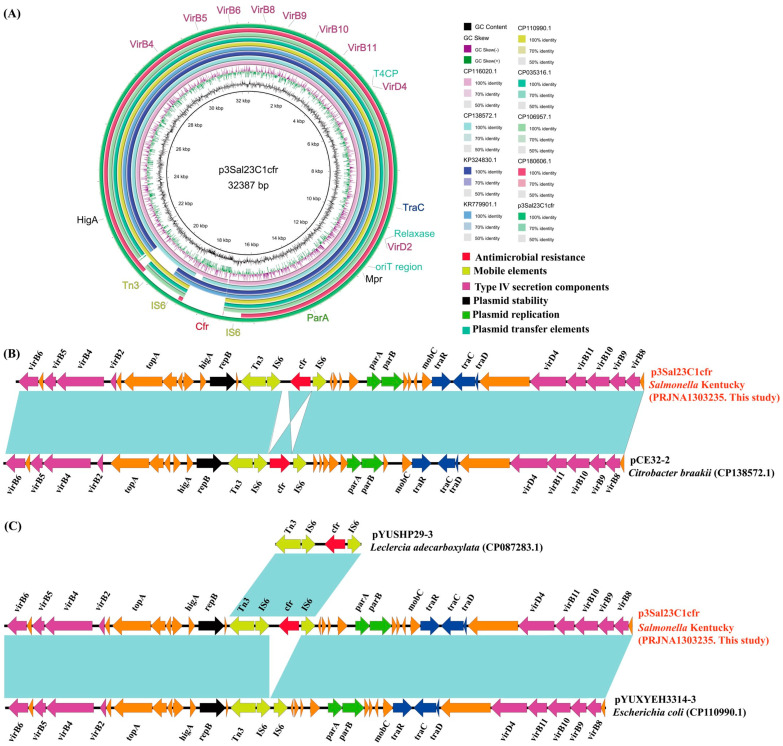
Sequence alignment of plasmid p3Sal23C1cfr and the gene environment of the *cfr* gene. (**A**) Comparison of the circular plasmid sequence between plasmids p3Sal23C1cfr, KR779901.1, KP324830.1, CP106957.1, CP180606.1, CP110990.1, CP035316.1, CP138572.1 and CP116020.1 of *Escherichia coli* strains (*n* = 6), *Citrobacter braakii* strain (*n* = 1) and *Salmonella* strain (*n* = 1). (**B**) Linear comparison of the plasmid sequences of *Salmonella* p3Sal23C1cfr and *Citrobacter braakii* (CP138572.1) pCE32-2. (**C**) Linear comparison of the plasmid sequences between plasmids p3Sal23C1cfr (*Salmonella*), pYUSHP29-3 (*Leclercia adecarboxylata*, CP087283.1) and pYUXYEH3314-3 (*Escherichia coli*, CP110990.1). Open arrows indicate coding sequences.

**Figure 4 foods-14-03025-f004:**
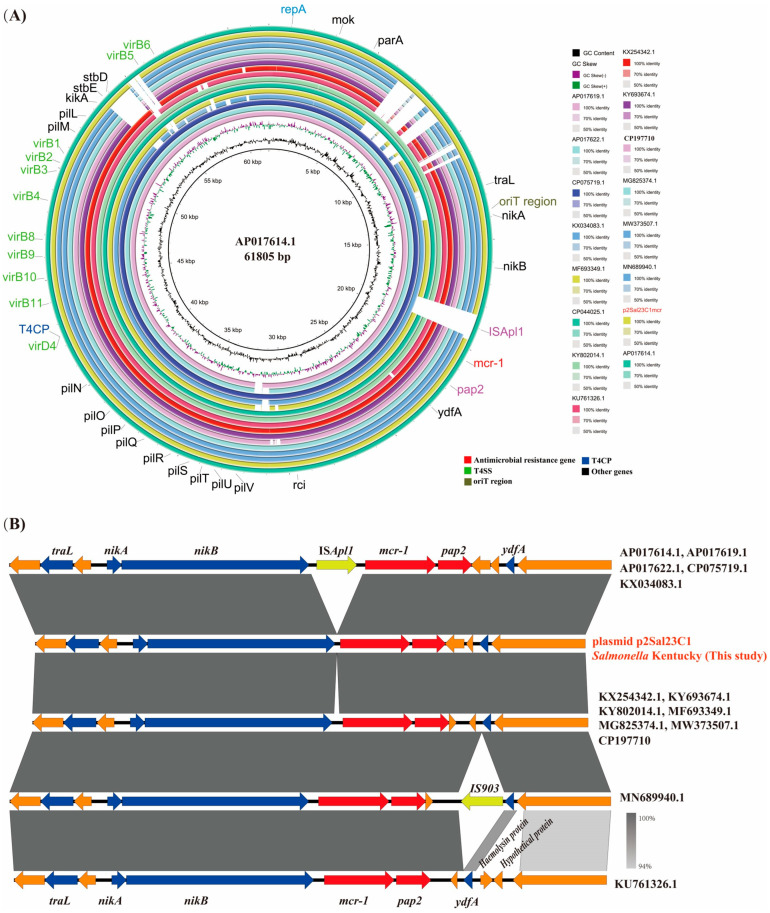
Comparative analysis of complete *mcr-1*-positive IncI2 plasmids. (**A**) Comparative visualization of 16 complete *mcr-1*-positive IncI2 plasmids using BRIG. The outermost ring depicts the plasmid AP017614.1 as the reference. Genes are color-coded by function. Horizontal transfer-associated genes are indicated: *T4CP* (blue) and *T4SS* clusters (green). The antimicrobial resistance gene *mcr-1* is shown in red. (**B**) Comparison of the *mcr-1* genetic context across all plasmids. All plasmids carry the *pap2-mcr-1* unit. Notably, plasmids AP017614.1, AP017619.1, AP017622.1, CP075719.1 and KX034083.1 contain the transposon IS*Apl1* upstream of this unit, while the remainder lack IS*Apl1*.

**Table 1 foods-14-03025-t001:** Antimicrobial susceptibility, conjugation rate and whole genome analysis of isolate Sal23C1 collected in this study.

Category	Antimicrobial Class	Antimicrobial Agent	MIC (mg/L)	Related Genes	Interpretation
Antimicrobial susceptibility testing	Folate pathway inhibitors	Sulfisoxazole	>2048	*sul1*	R
		Trimethoprim–sulfamethoxazole	>16/304		R
	Quinolones	Nalidixic acid	>128	*qnrS1*, *gyrA*(S83F, D87N), *parC*(S80I)	R
		Ofloxacin	16		R
		Ciprofloxacin	>8		R
	β-Lactam	Ampicillin	>128	*bla*_CTX-M-55_, *bla*_TEM-1B_	R
		Amoxicillin–clavulanic acid	>128/64		R
		Cefotaxime	16		R
		Cefepime	64		R
	Tetracyclines	Tetracycline	128	*tet*(A) *variant*	R
		Tigecycline	4		R
	Phenicols	Chloramphenicol	>128	*cfr, floR*	R
		florfenicol	>128		R
	Aminoglycosides	Streptomycin	256	*rmtB, aac(3)-IId, aac(6′)-Iaa, aadA7, aadA17, aph(3′)-Ia*	R
		Gentamicin	64		R
		Amikacin	256		R
		Kanamycin	256		R
	Rifamycins	Rifampicin	64	*arr-2*	R
	Polymyxins	Colistin	4	*mcr-1*	R
	Macrolides	Azithromycin	64	*mph*(A)	R
	Fosfomycins	Fosfomycin	512	*fosA3*	R
	Carbapenems	Meropenem	0.25		S
Collection time			2022		
Serotype			*Salmonella* Kentucky		
Sequence type			ST198		
*cfr* location			p3Sal23C1cfr (32,387 bp)		
*mcr-1* location			IncI2 (p2Sal23C1mcr, 63,103 bp)		
Conjugation rate			1.05 × 10^−6^~1.01 × 10^−8^		

Note: R, resistant; S, susceptible.

## Data Availability

The original contributions presented in this study are included in the article or supplementary material. Further inquiries can be directed to the corresponding author.

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
