# Peer review of "Characterization of an Extensively Drug-Resistant Salmonella Kentucky ST198 Co-Harboring cfr, mcr-1 and tet(A) Variant from Retail Chicken Meat in Shanghai, China"

_foods, 2025, doi:10.3390/foods14173025_

Round 1
Reviewer 1 Report
Comments and Suggestions for Authors
The manuscript submitted by Zhan et al. describes the complete genome of an XDR Salmonella strain isolated from a chicken meat sample. According to the WHO, third- and fourth-generation cephalosporin-resistant Enterobacteriales are a priority for AMR monitoring, which justifies the relevance of this scientific finding. Furthermore, the AMR-related genes are located on plasmids, demonstrating the potential for AMR dissemination through horizontal gene transfer to other strains.
However, the authors evaluated antimicrobial resistance (by MIC, disk diffusion, and WGS) from a single isolate, which considerably reduces the merit of the study. I believe the research would be much more interesting if the authors had presented at least the phenotypic AMR results of all isolates to provide an overall picture of AMR. In addition, reporting the frequency of gene detection by PCR during screening would greatly enrich the study. I suggest that the authors include these results in the manuscript to enhance its relevance, as in its current form, it has limited merit since it describes only a single isolate without further scientific contributions.
Revisions
- In the keywords, I suggest replacing words already present in the title with others that contextualize the study area for better manuscript indexing.
- In section 2.1, the authors need to provide a better description of the isolates. This should include the number of isolates per type of food analyzed and the isolation method applied (at least with a methodological reference). Maybe a supplementary file can be useful.
- In section 2.2, include the DNA extraction methodology and the quality assessment procedure.
- Did the authors not perform PCR to confirm species identification? Was it carried out only using API 20E test 93 strips?
- Section 2.2 should include the concentrations of the antimicrobials as well as the manufacturers of the disks.
- In section 2.3, clarify whether the MIC was determined with the same antimicrobials tested in the disk diffusion assay.
- In section 2.4, include the links to all websites used for the analyses.
- In section 2.6, include a supplementary file with the characteristics and accession numbers of all isolates
- The authors present the MIC results but do not mention whether all of them were consistent with the disk diffusion results, which are mentioned in the methodology but not included in the results section.
Author Response
Reviewer 1
Comments to the Author
Comments and Suggestions for Authors
Question 1: The manuscript submitted by Zhan et al. describes the complete genome of an XDR Salmonella strain isolated from a chicken meat sample. According to the WHO, third- and fourth-generation cephalosporin-resistant Enterobacteriales are a priority for AMR monitoring, which justifies the relevance of this scientific finding. Furthermore, the AMR-related genes are located on plasmids, demonstrating the potential for AMR dissemination through horizontal gene transfer to other strains.
However, the authors evaluated antimicrobial resistance (by MIC, disk diffusion, and WGS) from a single isolate, which considerably reduces the merit of the study. I believe the research would be much more interesting if the authors had presented at least the phenotypic AMR results of all isolates to provide an overall picture of AMR. In addition, reporting the frequency of gene detection by PCR during screening would greatly enrich the study. I suggest that the authors include these results in the manuscript to enhance its relevance, as in its current form, it has limited merit since it describes only a single isolate without further scientific contributions.
Response: Thank you very much for the positive comments, helpful suggestions and detailed corrections. We have reviewed the comments carefully and have revised the manuscript as your suggestions.
Regarding the phenotypic AMR results of all isolates, we sincerely thank the reviewer for this insightful comment. We fully agree that presenting the phenotypic antimicrobial resistance profiles of all 502 Salmonella isolates would provide a more comprehensive overview of the antimicrobial resistance situation in retail foods. However, due to the substantial number of isolates and the significant resources required for full antimicrobial susceptibility testing (including the labor-intensive procedures, preparation of multiple antibiotics and Mueller-Hinton agar plates), we focused our phenotypic antimicrobial resistance characterization on the isolate that co-harbored the three critical resistance genes (cfr, mcr-1, and tet(A) variant), which was the primary objective of this study. Nevertheless, to partially address the reviewer’s concern, we would like to note that all 502 isolates were initially screened by PCR for the presence of these three key resistance genes. Among them, only one isolate (Sal23C1) was positive for all three, underscoring the rarity and significance of this particular strain. While we did not perform full phenotypic antimicrobial susceptibility testing on all isolates, the genotypic screening itself provides valuable insight into the prevalence of these high-priority resistance determinants in the sampled population. We acknowledge that broader phenotypic antimicrobial resistance data would enhance the study, and we plan to incorporate more extensive antimicrobial susceptibility testing profiling in future surveillance projects when additional resources are available. We appreciate the reviewer’s suggestion and will certainly consider it in the design of our subsequent research.
Regarding the frequency of gene detection by PCR, we thank you for suggesting that reporting the frequency of gene detection would enrich our study. During PCR screening of the 502 isolates, we found that the tet(A) variant was the most prevalent, detected in 124 isolates, followed by mcr-1, which was present in 7 isolates. In contrast, the cfr gene was identified in only one isolate, indicating it is relatively uncommon in this population. Of particular concern, one isolate was found to co-harbor all three genes—cfr, mcr-1, and the tet(A) variant. We have added the relevant sentence “PCR screening of 502 Salmonella isolates from retail foods revealed varied prevalence of key resistance genes. The tet(A) variant was the most prevalent, detected in 124 isolates (24.7%), followed by mcr-1, which was present in 7 isolates (1.4%). The cfr gene was the rarest, identified in only one isolate (0.2%). Notably, a single isolate (0.2%), designated Sal23C1 and collected from retail chicken meat in Shanghai in 2022, was found to co-harbor all three genes—cfr, mcr-1 and the tet(A) variant.” in Line 216-221.
Revisions
Question 2: In the keywords, I suggest replacing words already present in the title with others that contextualize the study area for better manuscript indexing.
Response: We sincerely thank you for this valuable suggestion. As recommended, we have revised the keywords to eliminate redundancy with the title and to better reflect the study’s novel findings. The updated keyword list is now: “XDR Salmonella; Antimicrobial resistance genes; cfr; Whole genome sequencing; Conjugative plasmid; Mobile genetic elements; Poultry meat” in Line 35-36.
Question 3: In section 2.1, the authors need to provide a better description of the isolates. This should include the number of isolates per type of food analyzed and the isolation method applied (at least with a methodological reference). Maybe a supplementary file can be useful.
Response: Thank you very much for your valuable suggestion. We agree that a more detailed description of the isolates, including the number of isolates per food type and the isolation method used, would improve the clarity of the manuscript.
Accordingly, we have added the relevant sentence “The food types included pork (n = 170), poultry (n = 163), beef (n = 60), lamb (n = 40), shrimp (n = 15), roast meat (n = 12), freshwater fish (n = 9), steamed buns (n = 8), frozen dumplings (n = 6), sushi (n = 5), fried rice (n = 2), and cold vegetable dishes (n = 2). Salmonella isolation was performed using a modified adaptation of the ISO 6579-1:2017 protocol. Briefly, each sample was aseptically collected, placed in a sterile bag, and transported to the laboratory under refrigeration (4–8°C) within 8 hours. For processing, 25 g of each sample was homogenized with 225 mL of buffered peptone water (Huankai, China) and incubated at 37 ± 1°C with shaking (160 rpm) for 4 hours. Then, 1 mL of the pre-enrichment culture was transferred to 9 mL of tetrathionate broth (Huankai, China) and incubated at 42°C. After enrichment, a loopful of culture was streaked onto xylose lysine tergitol 4 agar (XLT-4; Huankai, China). Presumptive Salmonella colonies were subcultured on CHROMagar™ Salmonella plates (CHROMagar, France) and incubated at 37 °C for 24 hours. Salmonella identification was initially based on key biochemical characteristics, including H₂S production, anaerobic glucose fermentation without gas production, and urease negativity. Further confirmation was conducted using API 20E test strips (bioMérieux, France), a commercial identification system for Enterobacteriaceae. The verified isolates were subsequently subcultured on Luria-Bertani agar and genetically confirmed through PCR targeting the invA gene with specific primers (invA-F: CTTGATTGAAGCCGATGCCG/invA-R: TCATCGCACCGTCAAAGGAA). The identification of serotype was performed through standardized biochemical assays and commercial antisera (Statens Serum Institute, Denmark) following manufacturer protocols.” in Lines 97–116 as recommended.
Question 4: In section 2.2, include the DNA extraction methodology and the quality assessment procedure.
Response: Thank you very much for your valuable suggestion. We have added the relevant sentence “Total genomic DNA from overnight cultures of Salmonella was isolated with the TIANamp Bacteria DNA Kit (Tiangen Biotech, Beijing, China). In brief, bacterial cells from 1–2 mL culture was pelleted by centrifugation (10,000 rpm, 2 min). The pellet was resuspended in 200 μL Buffer GA, treated with 20 μL Proteinase K and 220 μL Buffer GB, mixed, and incubated at 70 °C for 10 min until lysate clarification. After adding 220 μL ethanol, the mixture was transferred to a CB3 spin column and centrifuged (10,000 rpm, 30 sec). The column was washed once with 500 μL Buffer GD and twice with 600 μL Buffer PW, with each wash step followed by a 30 sec spin. After a final 2 min centrifugation and air-drying for 2–5 min, DNA was eluted using 50–100 μL Buffer TE. DNA quality and concentration were determined via NanoDrop (Thermo Fisher Scientific).” in Lines 118–126 as recommended.
Question 5: Did the authors not perform PCR to confirm species identification? Was it carried out only using API 20E test 93 strips?
Response: Thank you for raising this important point regarding the confirmatory identification of the Salmonella isolates. We appreciate your thorough review. In addition to using API 20E test strips for definitive species identification, we also employed CHROMagar™ Salmonella plates (CHROMagar, France), assessed key biochemical characteristics, performed PCR targeting the invA gene, and used commercial antisera for serological confirmation. We have added the relevant sentence “Presumptive Salmonella colonies were subcultured on CHROMagar™ Salmonella plates (CHROMagar, France) and incubated at 37 °C for 24 hours. Salmonella identification was initially based on key biochemical characteristics, including H₂S production, anaerobic glucose fermentation without gas production, and urease negativity. Further confirmation was conducted using API 20E test strips (bioMérieux, France), a commercial identification system for Enterobacteriaceae. The verified isolates were subsequently subcultured on Luria-Bertani agar and genetically confirmed through PCR targeting the invA gene with specific primers (invA-F: CTTGATTGAAGCCGATGCCG, invA-R: TCATCGCACCGTCAAAGGAA). The identification of serotype was performed through standardized biochemical assays and commercial antisera (Statens Serum Institute, Denmark) following manufacturer protocols” in Lines 107–116 as recommended.
Question 6: Section 2.2 should include the concentrations of the antimicrobials as well as the manufacturers of the disks.
Response: Thank you for your valuable suggestion. We appreciate your comment regarding the inclusion of antimicrobial concentrations and disk manufacturers. In our study, antimicrobial susceptibility testing was performed using agar dilution and broth microdilution methods rather than disk diffusion. Therefore, disks were not used. The concentrations of the antimicrobials tested are provided in the manuscript, and all antibiotics were purchased from Sigma-Aldrich (USA). We have added the relevant sentence “The agar dilution method was applied to the following antibiotics at the indicated concentration ranges: sulfisoxazole (16–2048 mg/L), trimethoprim-sulfamethoxazole (0.5/9.5–16/304 mg/L), nalidixic acid (8–128 mg/L), ofloxacin (0.5–32 mg/L), ciprofloxacin (0.06–8 mg/L), ampicillin (4–128 mg/L), amoxicillin–clavulanic acid (4/2–128/64 mg/L), cefotaxime (1–64 mg/L), cefepime (1–64 mg/L), tetracycline (4–128 mg/L), chloramphenicol (8–128 mg/L), florfenicol (4–128 mg/L), streptomycin (8–512 mg/L), gentamicin (2–64 mg/L), amikacin (4–256 mg/L), kanamycin (8–256 mg/L), rifampin (4–128 mg/L), azithromycin (4–128 mg/L), fosfomycin (16–512 mg/L) and meropenem (0.06–16 mg/L). For tigecycline and colistin, MICs were determined using the broth microdilution method. All antibiotics used in this study purchased from Sigma-Aldrich (USA).” in Lines 140–156 as recommended.
Question 7: In section 2.3, clarify whether the MIC was determined with the same antimicrobials tested in the disk diffusion assay.
Response: Thank you for your careful reading and valuable comment regarding our methodology. In Section 2.3 of our manuscript, the Minimum Inhibitory Concentrations (MICs) for the Salmonella isolate Sal23C1 were determined exclusively using the agar dilution and broth microdilution methods, as recommended by the CLSI (2023) guidelines. The disk diffusion assay was not employed for any of the antimicrobial agents tested in this study.
I apologize for any confusion caused by the original wording in Section 2.3. To clarify: the MIC was determined by the two methods were used to test different antimicrobials based on standard guidelines. The broth microdilution method was used exclusively for tigecycline and colistin and the agar dilution method was used for the other antibiotics in the study. No antimicrobial agent was tested using both methods. The original text has been revised in the manuscript to make this distinction clearer. The relevant sentence “The agar dilution method was applied to the following antibiotics at the indicated concentration ranges: sulfisoxazole (16–2048 mg/L), trimethoprim-sulfamethoxazole (0.5/9.5–16/304 mg/L), nalidixic acid (8–128 mg/L), ofloxacin (0.5–32 mg/L), ciprofloxacin (0.06–8 mg/L), ampicillin (4–128 mg/L), amoxicillin–clavulanic acid (4/2–128/64 mg/L), cefotaxime (1–64 mg/L), cefepime (1–64 mg/L), tetracycline (4–128 mg/L), chloramphenicol (8–128 mg/L), florfenicol (4–128 mg/L), streptomycin (8–512 mg/L), gentamicin (2–64 mg/L), amikacin (4–256 mg/L), kanamycin (8–256 mg/L), rifampin (4–128 mg/L), azithromycin (4–128 mg/L), fosfomycin (16–512 mg/L) and meropenem (0.06–16 mg/L). For tigecycline and colistin, MICs were determined using the broth microdilution method. All antibiotics used in this study purchased from Sigma-Aldrich (USA).” was added in Lines 140–156.
Question 8: In section 2.4, include the links to all websites used for the analyses.
Response: Thank you for your valuable suggestion. We have modified the sentence to “Specifically, PlasmidFinder 2.1 (https://cge.food.dtu.dk/services/PlasmidFinder/) identified plasmid replicons, ResFinder 4.1 (https://cge.food.dtu.dk/services/ResFinder/) detected antimicrobial resistance genes, and MLST 2.0 (https://cge.food.dtu.dk/services/MLST/) determined the multi-locus sequence type (MLST). The serotype was predicted using SeqSero2 v1.2.1 (http://www.denglab.info/SeqSero2)” in Line 165-169.
Question 9: In section 2.6, include a supplementary file with the characteristics and accession numbers of all isolates
Response: We thank you for this constructive suggestion. As requested, we have now provided a supplementary file listing all isolates and plasmids used for the phylogenetic analysis in Section 2.6. This file (designated Supplementary File S1: ‘Isolates and plasmids for phylogenetic analysis). As recommended, we have added the relevant sentence “Detailed in Supplementary File S1 (‘Isolates and plasmids for phylogenetic analysis’) is a complete inventory of the bacterial isolates and plasmids employed for this study’s phylogenetic examination.” In Line 195-197.
Question 10: The authors present the MIC results but do not mention whether all of them were consistent with the disk diffusion results, which are mentioned in the methodology but not included in the results section.
Response: We thank the reviewer for this comment. In this study, antimicrobial susceptibility testing was conducted exclusively using agar dilution and broth microdilution methods in accordance with CLSI guidelines (2023), as stated in Section 2.3. Disk diffusion was not employed. The broth microdilution method was used exclusively for tigecycline and colistin and the agar dilution method was used for the other antibiotics in the study. No antimicrobial agent was tested using both methods. The original text has been revised in the manuscript to make this distinction clearer. The relevant sentence “The agar dilution method was applied to the following antibiotics at the indicated concentration ranges: sulfisoxazole (16–2048 mg/L), trimethoprim-sulfamethoxazole (0.5/9.5–16/304 mg/L), nalidixic acid (8–128 mg/L), ofloxacin (0.5–32 mg/L), ciprofloxacin (0.06–8 mg/L), ampicillin (4–128 mg/L), amoxicillin–clavulanic acid (4/2–128/64 mg/L), cefotaxime (1–64 mg/L), cefepime (1–64 mg/L), tetracycline (4–128 mg/L), chloramphenicol (8–128 mg/L), florfenicol (4–128 mg/L), streptomycin (8–512 mg/L), gentamicin (2–64 mg/L), amikacin (4–256 mg/L), kanamycin (8–256 mg/L), rifampin (4–128 mg/L), azithromycin (4–128 mg/L), fosfomycin (16–512 mg/L) and meropenem (0.06–16 mg/L). For tigecycline and colistin, MICs were determined using the broth microdilution method. All antibiotics used in this study purchased from Sigma-Aldrich (USA).” was added in Lines 140–156.

Reviewer 2 Report
Comments and Suggestions for Authors
The work provides crucial insights into the genetic basis of XDR in Salmonella Kentucky and the mechanisms facilitating the co-transfer of critical resistance genes through plasmids. This research adds significant value to the field of antimicrobial resistance surveillance and food safety.
Specific Suggestions for Improvement
1.Figures/Tables
Include a simplified comparative diagram or table illustrating the structure of MRR and SGI1-KI variant versus standard configurations.
- Methodology Clarification
State the number of replicates for the conjugation experiment and include variability measures (e.g., standard deviation).
Provide sequencing quality metrics (e.g., coverage, N50).
3.Discussion Enhancement
Expand discussion on potential intervention strategies in poultry production (e.g., antibiotic stewardship, biosecurity measures).
Add comments on possible fitness trade-offs for isolates harboring multiple resistance determinants.
Author Response
Reviewer 2
Comments to the Author
The work provides crucial insights into the genetic basis of XDR in Salmonella Kentucky and the mechanisms facilitating the co-transfer of critical resistance genes through plasmids. This research adds significant value to the field of antimicrobial resistance surveillance and food safety.
Response: Thank you very much for the positive comments, helpful suggestions and detailed corrections. We have reviewed the comments carefully and have revised the manuscript as your suggestions.
Specific Suggestions for Improvement
Figures/Tables.
Question 1. Include a simplified comparative diagram or table illustrating the structure of MRR and SGI1-KI variant versus standard configurations.
Response: Thank you for your suggestion to include a comparative illustration of the MRR and SGI1-KI variant structures. We have already addressed this in our manuscript. Figure 2 provides a detailed side-by-side comparison of the genetic architectures for both the Multidrug Resistance Region (MRR) and the Salmonella Genomic Island 1 (SGI1) variant found in our S. Kentucky ST198 isolate, against their standard reference configurations.
- Figure 2A illustrates the structure of the MRR inserted within the bcfH gene, contrasting it with a standard chromosomal context. The diagram uses color-coding to distinguish chromosomal genes (light blue), insertion sequences (IS, yellow), and antibiotic resistance genes (red).
- Figure 2B presents a comparative analysis of the SGI1-KI variant integrated downstream of the trmE gene, aligned with the canonical SGI1 reference sequence (AY463797). Resistance genes (red) and IS elements (yellow) are highlighted, and regions of sequence similarity are connected with grey shading to visualize homology.
We believe this figure effectively captures the key structural differences and innovations of these genomic elements. Please let us know if you require any further clarification or additional information regarding this figure.
Methodology Clarification
Question 2. State the number of replicates for the conjugation experiment and include variability measures (e.g., standard deviation).
Response: Thank you for this comment. As suggested, we have now explicitly stated the number of biological replicates and the measure of variability in the revised manuscript. As recommended, we have added the relevant sentence “We calculated the transfer frequency as the number of transconjugants per donor and expressed the conjugation efficiency as the mean ± standard deviation from three independent experiments.” in Line 184-186 and the sentence “The conjugation frequency was (1.15 ± 0.98) × 10⁻⁶.” in Line 480-481.
Question 3. Provide sequencing quality metrics (e.g., coverage, N50).
Response: Thank you for this important question. As recommended, we have added the relevant sentence “The raw sequence reads from the Salmonella isolate Sal23C1 were assembled de novo using the PacBio RS II Analysis system via the HGAP assembler. Leveraging the hierarchical strategy of HGAP along with PacBio long-read sequencing, this approach enabled the production of highly accurate and contiguous genome assemblies—essential for achieving complete genomic sequences.” in Line 160-163 and the sentence “The assembly of the total reads resulted in four contigs with a total length of 5,079,104 bp and an N50 of 4,889,727, achieving a genome coverage of 250×.” in Line 230-231.
Discussion Enhancement
Question 4. Expand discussion on potential intervention strategies in poultry production (e.g., antibiotic stewardship, biosecurity measures).
Response: Thank you very much for this excellent suggestion. We agree that expanding the discussion on intervention strategies is crucial for highlighting the practical implications of our research. Your comment has helped us significantly improve the translational impact of our manuscript. We have added the relevant sentence “The emergence of colistin-tigecycline-XDR S. Kentucky ST198 in retail chicken meat highlights the critical need for comprehensive intervention strategies within poultry production systems to curb the dissemination of antimicrobial resistance. The extensive arsenal of ARGs, such as mcr-1 and tet(A) variant, grants strains like Sal23C1 their challenging resistance profile. Alarmingly, antibiotic candidates under development for WHO critical-priority pathogens show concerning cross-resistance rates with existing agents, particularly against such isolates[37]. This underscores the urgent need to elucidate the genomic characteristics of these strains and understand the mechanisms enabling such devastating resistance combinations. To address this threat, antibiotic stewardship programs must be rigorously implemented to reduce the non-therapeutic use of critically important antibiotics, including colistin and tigecycline, in animal husbandry. Restrictions on these agents could mitigate the selective pressure driving the acquisition and spread of resistance genes. Enhanced biosecurity measures are also essential to prevent the introduction and transmission of resistant pathogens within farms. These include strict sanitation protocols, controlled access to poultry houses, and the use of protective clothing and equipment to minimize cross-contamination. Vaccination programs against Salmonella and the use of probiotics or prebiotics to promote gut health may further reduce reliance on antibiotics. Furthermore, continuous surveillance of antimicrobial resistance in both animals and food products, coupled with whole-genome sequencing, is vital to track resistance trends and identify emerging threats early. The integration of these strategies within a One Health framework is crucial to effectively combat the spread of XDR pathogens from poultry to humans through the food chain.” in Line 271-290 and the sentence “In light of these findings, it is imperative to implement comprehensive intervention strategies within poultry production systems to curb the dissemination of such high-risk resistant bacteria. Antibiotic stewardship programs that enforce responsible use of antimicrobials—including restrictions on non-therapeutic applications and the prohibition of critically important antibiotics for growth promotion—are essential to reduce selective pressure for resistance development. Additionally, enhanced biosecurity measures should be adopted to prevent the introduction and spread of pathogens within farms. These include strict access control, sanitation protocols, effective waste management, and rodent and insect control. Integrated surveillance of antimicrobial resistance across the farm-to-fork continuum should be strengthened, coupled with periodic training of farmers and veterinarians on appropriate antibiotic use and infection prevention practices. Without such multidimensional interventions, the continued emergence and spread of pan-resistant pathogens in the food chain may become increasingly common, posing severe risks to public health.” in Line 520-531.
Question 5. Add comments on possible fitness trade-offs for isolates harboring multiple resistance determinants.
Response: Thank you very much for this insightful and valuable suggestion. We completely agree that discussing the potential fitness trade-offs associated with harboring multiple antimicrobial resistance (AMR) determinants is a crucial aspect of understanding the ecology and evolution of resistant strains. Your comment has significantly improved the depth of our discussion. We have added the relevant sentence “The acquisition of extensive resistance determinants, particularly through large genomic insertions like the MRR that disrupt native genes (e.g., bcfH), can potentially impose a fitness cost on the bacterial host in the absence of antibiotic selection. However, the global dissemination and persistence of XDR S. Kentucky ST198 clones suggest that any such costs are effectively mitigated. The chromosomal integration of key resistance genes within stable genetic platforms (MRR, SGI1-KI) enhances their persistence compared to plasmid-borne genes, which may be lost more readily. Furthermore, the observed genomic rearrangements and deletions may themselves represent compensatory evolution that reduces the metabolic burden. The formidable survival advantage provided by multidrug resistance in environments with high antibiotic selective pressure appears to far outweigh any potential fitness defects, enabling the successful clonal expansion and spread of this high-risk lineage with minimal trade-offs.” in Line 387-398.

Round 2
Reviewer 1 Report
Comments and Suggestions for Authors
All my suggestions were adressed.